# Endothelial and Vascular Health: A Tale of Honey, H_2_O_2_ and Calcium

**DOI:** 10.3390/cells10051071

**Published:** 2021-04-30

**Authors:** Elia Ranzato, Gregorio Bonsignore, Mauro Patrone, Simona Martinotti

**Affiliations:** 1DiSIT—Dipartimento di Scienze e Innovazione Tecnologica, University of Piemonte Orientale, Viale Teresa Michel 11, 15121 Alessandria, Italy; elia.ranzato@uniupo.it (E.R.); gregorio.bonsignore@uniupo.it (G.B.); mauro.patrone@uniupo.it (M.P.); 2DiSIT—Dipartimento di Scienze e Innovazione Tecnologica, University of Piemonte Orientale, Piazza Sant’Eusebio 5, 13100 Vercelli, Italy

**Keywords:** buckwheat honey, endothelial cells, intracellular calcium, hydrogen peroxide, wound healing

## Abstract

Intracellular Ca^2+^ regulation plays a pivotal role in endothelial biology as well as during endothelial restoration processes. Interest in honey utilization in wound approaches is rising in recent years. In order to evaluate the positive effects of buckwheat honey on endothelial responses, we utilized an immortalized endothelial cell line to evaluate cellular responses upon honey exposure, with particular interest in Ca^2+^ signaling involvement. The results highlight the positive effects of buckwheat honey on endothelial cells’ responses and the central role played by Ca^2+^ signaling as an encouraging target for more efficacious clinical treatments.

## 1. Introduction

The integrity of the endothelial monolayer may be compromised by either disturbed blood flow or pathological conditions. The prolonged and/or repeated exposure of the endothelial monolayer to well-known cardiovascular risk factors can induce a dramatic impact on vascular endothelium. The injured endothelium is regarded as the early event that results in the onset of severe vascular disorders, including atherosclerosis, acute myocardial infarction, brain stroke, and aortic aneurysm.

It is, therefore, not surprising that novel approaches are crucial for clinicians to achieve the early restoration of a fully competent endothelium in order to gain more effective long-term results after vascular regenerative surgery [1,2].

Intracellular Ca^2+^ signals play a central role in controlling endothelial survival, proliferation, motility, and differentiation, maintaining substantially conserved mechanisms even through the vascular tree variability [3]. Therefore, it should not come as a surprise that there are variations in the calcium profile in response to physical stimuli, exogenous agonists (growth factors and hormones), natural compounds or second messengers [4]. These kinds of stimuli, acting on the extracellular side of the plasma membrane, determine a reversible increase in the intracellular calcium concentration ([Ca^2+^]_i_), usually taken back to the resting conditions after agonist removal or even in its presence, thanks to the active extrusion from the cytoplasm [5]. The [Ca^2+^]_i_ increase arises from the entry of extracellular Ca^2+^ through plasma membrane Ca^2+^-permeable ion channels and by the release of Ca^2+^ from the endoplasmic reticulum (ER), the main intracellular Ca^2+^ store. Calcium release from ER is normally due to the rise in cytosolic levels of inositol-1,4,5-trisphosphate (InsP_3_), a diffusible and ubiquitous intracellular second messenger [4].

An attractive natural product that could be used to achieve an efficient rebuilding of the endothelium is honey, a beehive product already known and used in days gone by [6]. Honeys, in particular dark honeys, are characterized by an intrinsic production of hydrogen peroxide (H_2_O_2_) that confers the capability to the honey to begin the wound healing process and trigger the [Ca^2+^]_i_ mobilization [7]. 

The aim of this study was to determine the ability of buckwheat honey (genus *Fagopyrum*), a polyphenol-rich dark honey known for its anti-inflammatory properties [8], to stimulate wound closure in endothelial cells, investigating the signal cascade activation in more depth.

## 2. Materials and Methods

### 2.1. Honey Samples

Buckwheat honey (BH) was obtained from Yamada Apiculture Center, Inc., Tomata-Gun, Okayama (Japan).

### 2.2. Cell Culture and Reagents

Except when otherwise specified, reagents were all purchased from Merck (previously Sigma-Aldrich, Milan, Italy). The bEND5 cell line (American Type Culture Collection, Manassas, VA, USA), an immortalized mouse cell line from brain capillary endothelial cells, was maintained at 37 °C, 5% CO_2_ in a complete medium (DMEM (high glucose, 4.5 g/L), 10% FBS, L-glutamine (200 mM, 100 U/mL penicillin and 100 mg/mL streptomycin) [9].

### 2.3. Scratch Wound Test

Confluent bEND5 monolayers were scratched with a sterile 2–20 µL pipette tip [10]. Next, detached cells were washed away, and cultures were rinsed for 24 h with treating solution (0.1–1% BH in complete medium). Afterward, cells were fixed in 3.7% formaldehyde in PBS for 30 min, then stained with 0.1% toluidine blue at room temperature for 15 min. The distance between the edges of the wound space was measured at T0 (at wounding) and at T24 (end of exposure) by using an inverted microscope (Leica Microsystems) equipped with a digital camera. Digitized pictures of wounds were analyzed using the NIH ImageJ software. Wound closure rate was calculated as the difference between wound width at T0 and at T24. 

### 2.4. Measurements of Free Cytosolic Ca^2+^ Concentration ([Ca^2+^]_i_)

bEND5 cells were allowed to settle overnight on glass-base dishes (Iwaki Glass, Inc., Tokyo, Japan), then loaded in the dark at 37 °C for 30 min with 20 µM Fluo-3/AM, a cell-permeant, fluorescent calcium probe. The loading buffer was composed as described in previous work [11]. After probe loading and washing, cells were observed by confocal time-lapse analysis, using a Zeiss LSM 510 confocal system interfaced with a Zeiss Axiovert 100 m microscope (Carl Zeiss Inc., Oberkochen, Germany). 

Excitation was obtained by the 488 nm line of an Ar laser, and the emission was collected using a 505–550 bandpass filter. The laser power was reduced to 1% in order to lower probe bleaching.

Confocal imaging was performed with a resolution of 512 × 512 pixels at 256 intensity values, with a framing rate of 1 frame/5 s. 

Several cells were viewed together through a 20 × Plan-Neofluar Zeiss objective (0.5 NA). Fluo-3/AM fluorescence was measured in digitized images as the average value over defined contours of individual cells, using the ROI-mean tool of the Zeiss LSM 510 2.01 software. Fluo-3 calibration was achieved by the following equation [12]
Ca^2+^ = Kd(F − Fmin)/(Fmax − F)
where Kd = 400 nmol/L. 

Fmax and Fmin relate to maximum and minimum fluorescence intensities obtained by Fluo-3/AM calibration after the cell exposure to 500 μm A23187 for about 10 min, followed by the addition of 20 mm EDTA.

### 2.5. RNA Interference (siRNA)

RNAi for AQP3 was performed with an N-ter Nanoparticle siRNA Transfection System, which transfected cells with 5 µm siRNA oligonucleotides (Table 1) or with equimolar scramble siRNA, by using commercial non-targeting siRNA (MISSION siRNA Universal Negative Control).

### 2.6. Statistical Analysis

Statistics were achieved with GraphPad Prism 8 (Graphpad Software Inc., GraphPad Software, Inc, San Diego, CA, USA). Depending on the data, one-way or two-way ANOVAs were calculated, and the required corrections (Tukey’s test, Bonferroni correction, Dunnett’s post-test and Siddak test) were successively applied. For each experiment, statistical details (test used, value of *n*, replicates, *p* value, etc.) can be found in the Figure Legends.

## 3. Results

### 3.1. Cells Viability Assay

After 24 h treatment with different concentrations of buckwheat honey (BH; 0–50% *w*/*v*), bEND5 cells showed a dose-dependent effect with EC_50_ and EC_05_ values, as in Table 2.

### 3.2. Scratch Wound Assay

The BH was used at 0.1%, 0.5% and 1% to treat bEND5 cells to perform the scratch wound test. After 24 h of treatment, BH boosted a healing effect 2.5–3 times higher than the control condition (Figure 1).

### 3.3. Evaluation of Free Cytosolic Ca^2+^ Concentration ([Ca^2+^]_I_) Variation 

To evaluate the variation in [Ca^2+^]_i_ induced by BH, a time-lapse analysis was performed with confocal microscopy on cells previously incubated with the fluorescent probe Fluo-3-AM. By treating this with the increasing concentrations of honey (0.5–1–2% *w*/*v*), a dose-dependent response in terms of [Ca^2+^]_i_ variation was observed. Such a variation is described by a peak phase subsequent to the addition of honey, followed by a plateau phase leading again to homeostatic [Ca^2+^]_i_ (Figure 2). 

### 3.4. Effect of H_2_O_2_ on the [Ca^2+^]_i_ Variation

As previously known, the presence of honey in the extracellular medium leads to the production of H_2_O_2_ which influences and modifies intracellular behavior in keratinocytes [7].

Starting from this notion, it was therefore assessed whether the presence or not of H_2_O_2_ in the extracellular environment of endothelial cells could modulate the response of cells to BH treatment. The bEND5 cells were pretreated for 30 min with 1000 U of catalase, which is known to degrade H_2_O_2_. The absence of H_2_O_2_ outside of the cell resulted in a complete abolition of the peak detected after treatment with BH (Figure 3A). H_2_O_2_ extracellularly produced by BH could enter the cells through peroxyporins, and the contribution of AQP3 in keratinocytes has been already highlighted in this way [7].

Therefore, the observation was repeated under RNAi conditions for AQP3. The AQP3 deficiency determined the abrogation of the Ca^2+^ peak confirming the role of H_2_O_2_ in inducing an intracellular response after BH treatment and that the AQP3 needs to allow its entry into the cytoplasm (Figure 3B). To further highlight the importance of H_2_O_2_ as a trigger for the endothelial cell’s response, the scratch wound assay was performed by treating the scratched monolayer with 0.5% BH, but in the presence or not of catalase as previously indicated, or after RNAi for AQP3. The absence of H_2_O_2,_ or the lack of AQP3, determined the ineffectiveness of BH treatment (Figure 3C,D). To further support the involvement of H_2_O_2_, the variation in [Ca^2+^]_i_ after treatment with artificial honey was evaluated [7,11]. Artificial honey itself did not produce any effect, but when added with a quantity of H_2_O_2_ equivalent to that present in BH, a variation comparable to that obtained after treatment with 2% BH was recorded (Appendix A).

### 3.5. Extracellular Ca^2+^ Involvement

To define the origin of calcium involved in the intracellular variation due to BH treatment, the time-lapse observation was repeated by treating with 2% honey in the presence or not of extracellular Ca^2+^ (0Ca^2+^). The absence of extracellular Ca^2+^ resulted in a reduction of about 80% of the peak previously observed after the treatment with honey (Figure 4).

### 3.6. Involvement of the Calcium Toolkit in the [Ca^2+^]i Variation

It is already known that endothelial cells are characterized by a high expression of the Transient Receptor Potential cation channel, subfamily M, member 2 (TRPM2) channel which mediates Ca^2+^ entry from the extracellular environment through a redox dependent mechanism [13,14].

Moreover, we also previously defined in keratinocytes that honey is able to induce Ca^2+^ entry across the TRPM2 channel [7]. Based on this knowledge, a confocal microscopy analysis was performed focusing on this channel, and to disclose its involvement, the cells were pre-treated with 10 µM econazole (a TRPM2 inhibitor [15,16,17,18,19]). TRPM2 inhibition determined an almost complete reduction in the [Ca^2+^]_i_ increase caused by honey treatment (Figure 5).

Even in this condition, the peak detected after treatment with honey was lowered by about five times in a comparable way with that found in 0Ca^2+^ condition.

The addition of BH, both in the 0Ca^2+^ condition and in the presence of econazole, induces a little variation in the [Ca^2+^]_i_, significantly lower than the one determined by BH exposure, but statistically higher than the control condition. This variation could be due to a release from the endoplasmic reticulum (ER). Therefore, the observations were repeated in the presence of some inhibitors of the PLC-IP_3_ pathway: bEND5 was pre-treated with 50 µM 2-APB (inhibitor of the IP_3_-induced Ca^2+^ release [20,21,22,23]) or with 10 µM U73122 (a potent phospholipase C (PLC) inhibitor [24,25]). In the presence of both inhibitors, the peak induced by the treatment with BH underwent a significant decrease by about three times. These data were confirmed by pretreating cells with 10 mm caffeine which caused the inhibition of IP_3_R_3_-mediated Ca^2+^ release [26] (Figure 6).

### 3.7. Role of Ca^2+^ Toolkit in the Endothelial Wound Closure Response

Intracellular calcium variations are well known to generate the signal cascade that leads to wound healing [27,28], therefore, starting from the above [Ca^2+^]_i_ observations, the same inhibitors were utilized in a scratch wound assay. Furthermore, PD98059 [28], selective inhibitor of MAP kinase kinases (MAPKK), MEK1 and MEK2, was used. As in Figure 7A, all the inhibitors of the PLC-IP_3_ pathway significantly abolished the wound closure induction given by honey, as observed in the presence of a catalase, underlining the association between H_2_O_2_, intracellular calcium variations and MEK 1/2 pathways in the mechanism stimulated by honey. 

Inhibition of the extracellular Ca^2+^ entry by using econazole only decreases the wound healing rate (Figure 7B).

## 4. Discussion

Disturbed blood flow, pathological conditions or recognized cardiovascular risk factors (i.e., hyperlipidemia, hypercholesterolemia, hypertension, ageing and smoking) could weaken the structural integrity of the vascular endothelium, causing vascular injury [1]. Moreover, surgical procedures aimed at re-establishing blood flow could distress the endothelial integrity.

Therefore, there is a need for innovative approaches to hit a prompt renovation of the endothelium to improve any long-term effects after vascular regenerative surgery [1].

A possible novel strategy for the treatment of endothelial injury could be the use of honey, a natural product that in recent decades, as in ancient times, sparks interest for its several virtues and pharmacological properties [6,29,30,31,32,33].

In the present study, we investigated the potential effect of honey, in particular buckwheat honey (BH), on an endothelial cell line, bEND5, in terms of regenerative capability.

The first study, as already demonstrated on other cell lines, i.e., keratinocytes and fibroblasts [29,30], highlighted low cytotoxicity of BH on bEND5 cells. The treatment with BH, when performed on a scratched monolayer with an increasing concentration range (0.1–0.5–1% *w*/*v*), exhibited a positive effect determining a 2.5–3-fold increase in the wound closure rate. In particular, the concentration of 0.5% was the most effective. 

One of the principal factors in the wound healing process is the intracellular calcium [34], therefore we evaluated the variation in intracellular calcium concentration. 

We observed that this increase was dose-dependent, displaying the onset of a peak immediately after the addition of BH that came back to homeostatic values in a maximum of 200 s. 

As was already demonstrated, the treatment with honey determined the increase in H_2_O_2_ in the extracellular space and this amount of peroxide could enter inside the cells moving through a specific aquaglyceroporin, i.e., aquaporin 3 (AQP3). The entry of H_2_O_2_ in the cytoplasm triggers the intracellular signaling that operates through the wound healing process [7]. Such an aquaporin is expressed also in endothelial cells [35] and allows the entry of H_2_O_2_ that could start the signaling cascade. This statement, removing H_2_O_2_ from the extracellular environment or abolishing the activity of AQP3, was confirmed by the complete decrease in the calcium peak and the ineffectiveness of BH treatment in the induction of wound healing. This observation is interesting because some calcium channels are redox regulated and H_2_O_2_ could acts as a signaling molecule. 

Starting from the knowledge that H_2_O_2_ could activate and open some plasma membrane channels, allowing the extracellular calcium entry [13], to better define the dynamics correlated to the [Ca^2+^]_i_ variation, it was determined the origin of calcium. Under [0Ca^2+^]_i_ conditions, even with the presence of 2%BH, a peak 2.5–3-fold less than the one observed previously was recorded. This highlighted the extracellular calcium as the main contributor of the [Ca^2+^]_i_ variation. It is known from the literature that endothelial cells are characterized by a high expression and activity of the TRPM2 channel involved in a conspicuous entry of calcium into the cell [1,14].

The traces obtained after pretreatment with econazole are consistent with those observed in [0Ca^2+^], therefore it could be possible to identify a TRPM2 channel as the main access route of extracellular calcium after BH treatment.

Since the inhibition of extracellular calcium entry did not result in a complete reduction in the peak, it was decided to consider a possible role of the intracellular stores, focusing the attention on the endoplasmic reticulum (ER). As previously demonstrated on the same cells, calcium from ER plays a fundamental role in the response to stimuli that induces tissue regeneration, in particular the PLC-IP_3_ pathway [14]. The PLC inhibition by pretreating cells with U73122 [24] and InsP_3_R inhibition by using 2-APB [36] or caffeine [37], determined a reduction in the BH-induced [Ca^2+^]_i_ increase. 

Intriguingly, these observations regarding calcium dynamics, suggest the pivotal role of honey-produced H_2_O_2_ as a physiological mediator of cellular behavior in response to BH treatment. This critical role is due to the ability of extracellular H_2_O_2_ to activate a variety of nonreceptor- and receptor-type tyrosine kinase in heterogeneous cell systems, leading to a tyrosine kinase-dependent IP_3_ production and consequent calcium release, suggesting an activation of PLC via tyrosine phosphorylation by H_2_O_2_-activated tyrosine kinases [38]. The H_2_O_2_ mediated PLC activation drives the production of IP_3_ and diacylglycerol (DAG), and the latter activates protein kinase C (PKC) that starts the MAPK cascade ending with the activation of MEK1/2 and ERK1/2, fundamental actors in the wound healing mechanism [39]. In order to verify these statements, the scratch wound assay was performed by treating cells with BH, but in the presence of PLC-IP_3_ pathway inhibitors or PD98059, a selective inhibitor of MEK1 and MEK2 [40], and in all conditions the wound closure rate fell down to the control level. Instead, in the presence of econazole, there was only a reduction.

Taken together, these data suggest a mechanism that could be summarized as follows (Figure 8):Buckwheat honey treatment results in the production of H_2_O_2_ outside the cells,H_2_O_2_ enters the cells through AQP3,H_2_O_2_ in the cytoplasm starts two different ways of action to increase the [Ca^2+^]_i_:
o The activation of a TRPM2 channel determining calcium entry from the extracellular space,o The activation of a PLC-IP_3_ pathway leading to a calcium release from the ER,The [Ca^2+^]_i_ increase contributes to the activation of PLC,PLC activation begins the MAPK pathway that results in the enhancing of the wound closure rate.

This evidence hints at the central role played by Ca^2+^ signaling as an encouraging target headed for more efficacious clinical treatments. This suggestion is strengthened by in vitro evidence showing that regeneration may be boosted by modulating intracellular Ca^2+^ dynamics. 

Accordingly, the addition of honey, through H_2_O_2_ entry, stimulates wound closure in endothelial cells by inducing an increase in the [Ca^2+^]_i_, that, in turn, results in ERK1/2 activation.

## Figures and Tables

**Figure 1 cells-10-01071-f001:**
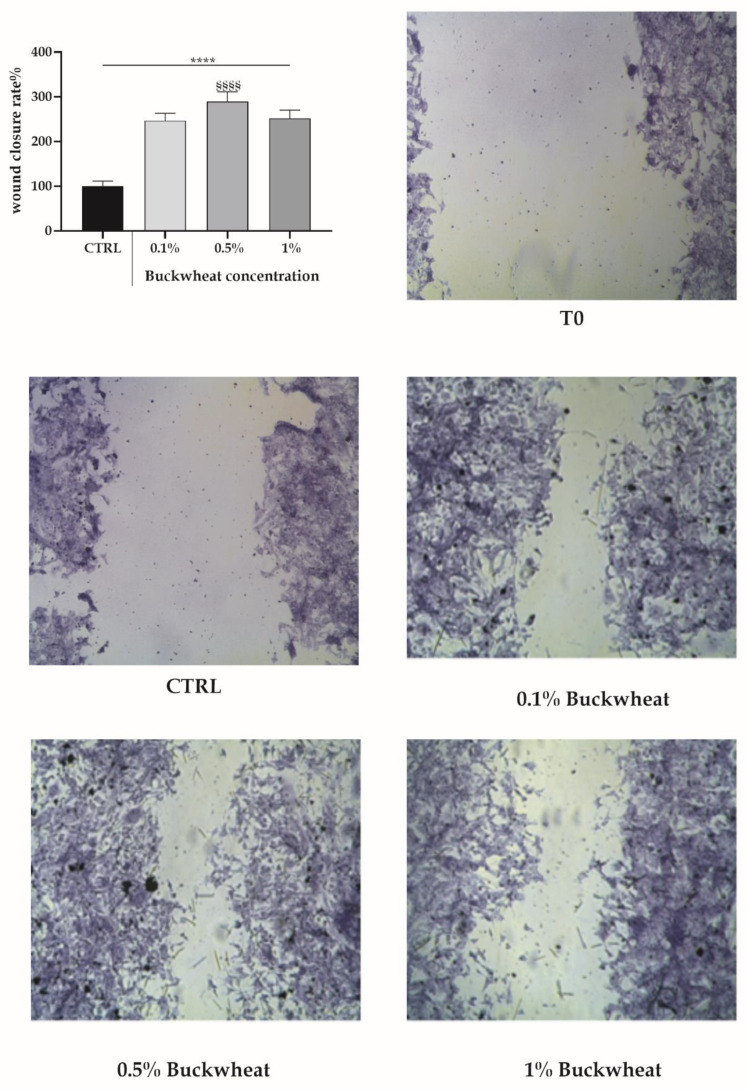
The scratch wound test in bEND5 cells exposed to BH. The micrographs of scratched bEND5 cells at T0 and when exposed to increasing concentrations of BH for 24 h. Insert. The closing percentage wound values after exposure for 24 h at increasing concentrations of the BH. Statistics on bars indicate differences with respect to control (CTRL) condition determined by a One-Way ANOVA followed by Dunnett’s test (**** *p* < 0.0001), or differences between BH treatments determined by a One-Way ANOVA followed by Tukey’s test (§§§§ *p* < 0.0001).

**Figure 2 cells-10-01071-f002:**
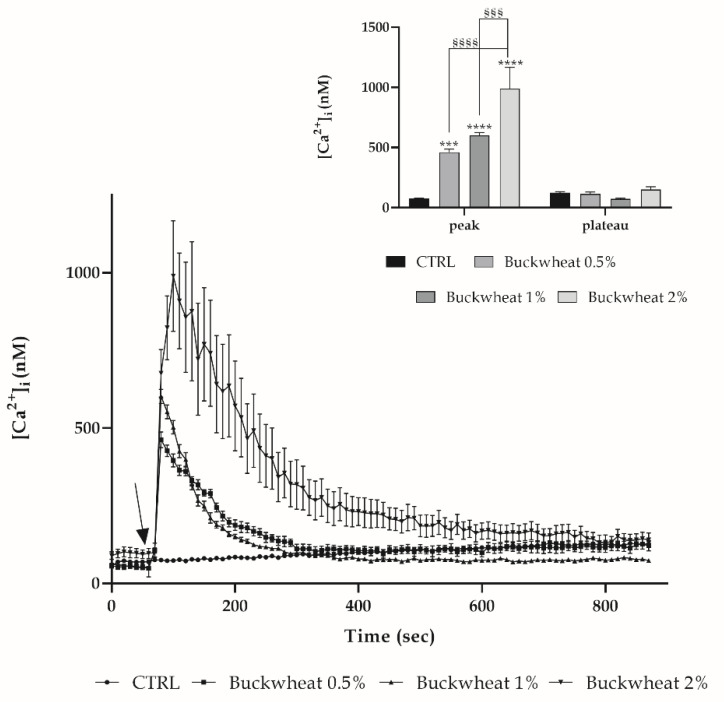
Dose-dependent variation of [Ca^2+^]_i_ induced by BH in bEND5 cells. [Ca^2+^]_i_ modifications recorded at 5 s intervals, exhibiting any alterations in control conditions, and dose-dependent increases subsequent to the treatment with different honey concentrations, i.e., 0.5%, 1% and 2% *w*/*v*. The arrow shows the BH addition after 60 s. Data are means ± s.e.m. of [Ca^2+^]_i_ traces were recorded in different cells. Number of cells: for each concentration, 40 cells from 3 exp. Insert. Mean ± s.e.m. of Ca^2+^ at the peak response. Number of cells are as before. Statistics on bars indicate differences with respect to CTRL determined by a Two-Way ANOVA followed by Tukey’s test (*** *p* < 0.001; **** *p* < 0.0001), or differences between BH treatments determined by a Two-Way ANOVA followed by Tukey’s test (§§§ *p* < 0.001; §§§§ *p* < 0.0001).

**Figure 3 cells-10-01071-f003:**
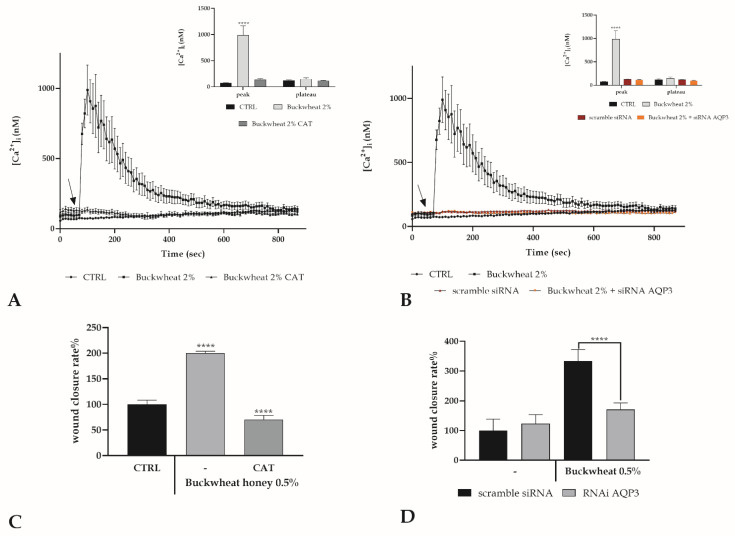
H_2_O_2_ involvement in the BH boosted Ca^2+^response. (**A**) The Ca^2+^ response to 2% BH is completely abolished by Catalase (CAT 1000U, 30 min pre-incubation). Data are means ± s.e.m. of [Ca^2+^]_i_ traces measured in different cells. The arrow shows the addition of BH after 60 s. Number of cells: BH: 40 cells from 3 exp; BH + CAT: 40 cells from 3 exp Insert. Mean ± s.e.m. of the Ca^2+^ peak response measured under the indicated treatments. Number of cells as before. Asterisks on bars indicate statistical differences determined by a Two-Way ANOVA followed by Tukey’s test (**** *p* < 0.0001). (**B**) BH-induced [Ca^2+^]_i_ variation was completely nullified in cells transfected with RNAi targeting AQP3. BH was added at 2% *w*/*v*. Data are means ± s.e.m. of [Ca^2+^]_i_ traces recorded in different cells. Number of cells: BH: 30 cells from 3 exp; scramble siRNA: 40 cells from 3 exp; BH after RNAi for AQP3: 45 cells from 3 exp. Insert. Mean ± s.e.m. of the Ca^2+^ peak response measured under the indicated treatments. Number of cells as before. Statistics as before. (**C**) Wound closure rate after 24 h exposure at 0.5% concentration of BH, in the presence or not of CAT 1000U. Asterisks on bars indicate statistical differences with respect to CTRL determined by a One-Way ANOVA followed by Dunnett’s test (**** *p* < 0.0001). (**D**) Wound closure rate after 24 h exposure at 0.5% concentration of BH, after scramble siRNA or RNAi for AQP3. Asterisks on bars indicate statistical differences determined between BH treatments by a One-way ANOVA followed by Bonferroni correction (**** *p* < 0.0001).

**Figure 4 cells-10-01071-f004:**
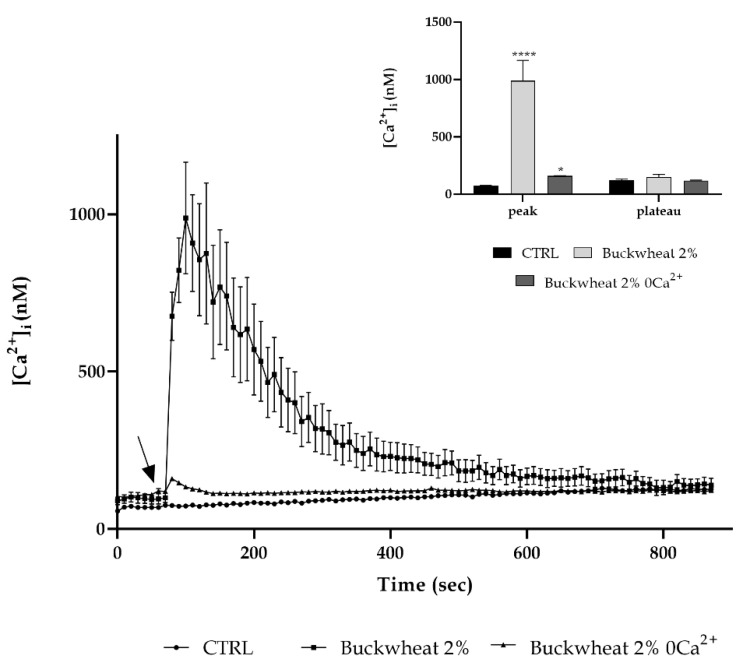
Extracellular Ca^2+^ entry is primarily involved in response to honey. The Ca^2+^ response to 2% BH was significantly reduced in the 0Ca^2+^ condition. The arrow shows the addition of honey after 60 s. Data are means ± s.e.m. of [Ca^2+^]_i_ traces recorded in different cells. Number of cells: buckwheat honey: 40 cells from 3 exp; buckwheat honey 0Ca^2+^: 30 cells from 3 exp. Insert. Mean ± s.e.m. of the Ca^2+^ peak response. Number of cells as before. Asterisks on bars indicate statistical differences with respect to CTRL determined by a Two-Way ANOVA followed by Siddak test (* *p* < 0.05; **** *p* < 0.0001).

**Figure 5 cells-10-01071-f005:**
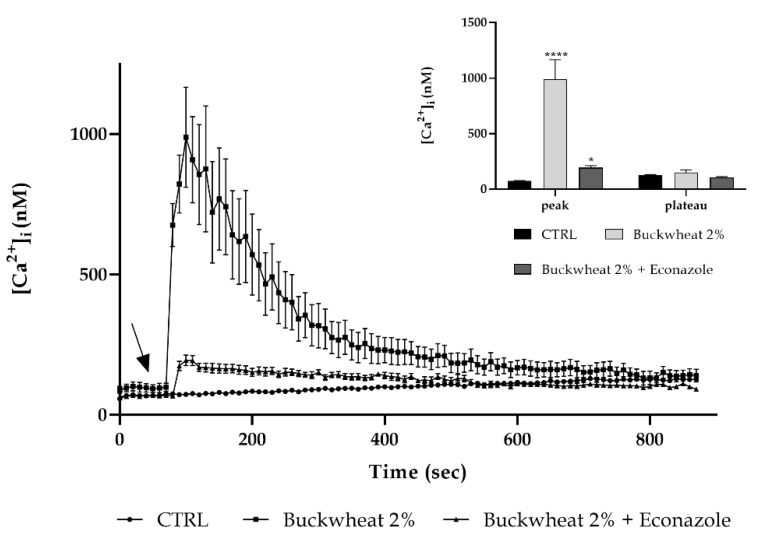
The TRPM2 channel contribution to the Ca^2+^ entry. The Ca^2+^ response to 2% BH was significantly reduced after pretreatment with econazole (preincubation time: 30 min). The arrow shows BH addiction after 60 s. Data are means ± s.e.m. of [Ca^2+^]_i_ traces recorded in different cells. Number of cells: BH honey: 40 cells from 3 exp; BH + Econazole: 50 cells from 3 exp. Insert. Mean ± s.e.m. of the Ca^2+^ peak response. Number of cells as before. Asterisks on bars specify statistical differences with respect to CTRL determined by a Two-Way ANOVA followed by Siddak test (* *p* < 0.05; **** *p* < 0.0001).

**Figure 6 cells-10-01071-f006:**
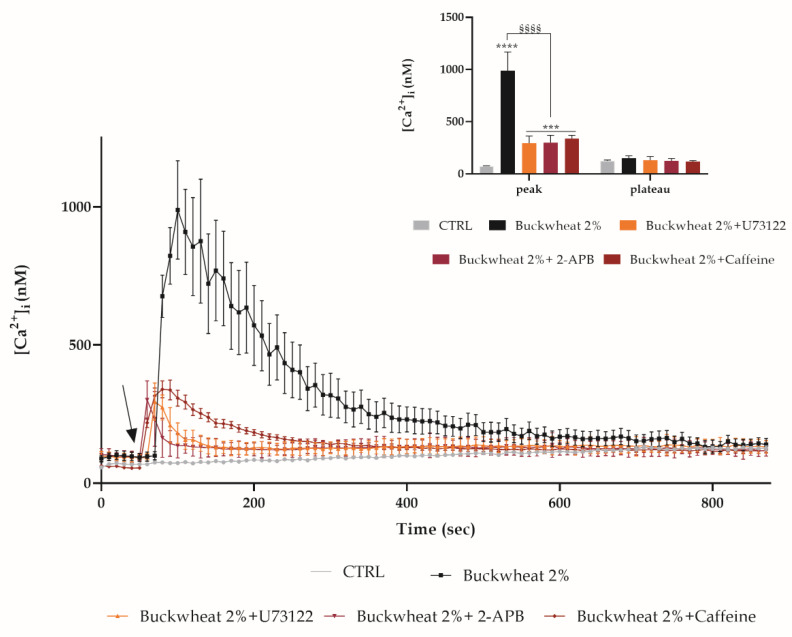
The PLC-IP_3_ pathway involvement. The Ca^2+^ response to 2% BH was significantly reduced in the presence of 2-APB, U73122, caffeine (30 min pre-incubation for each drug). The arrow shows BH addition after 60 s. Data are means ± s.e.m. of [Ca^2+^]_i_ traces recorded in different cells. Number of cells: BH honey: 40 cells from 3 exp; BH + 2-APB: 45 cells from 3 exp; BH + U73122: 45 cells from 3 exp; BH + caffeine: 45 cells from 3 exp. Insert. Mean ± s.e.m. of the Ca^2+^peak response. Number of cells as before. Asterisks on bars specify statistical differences with respect to CTRL determined by a Two-way ANOVA follow by Dunnett’s test (*** *p* < 0.001, **** *p* < 0.0001), or differences between BH treatments determined by a Two-Way ANOVA followed by Tukey’s test (§§§§ *p* < 0.0001).

**Figure 7 cells-10-01071-f007:**
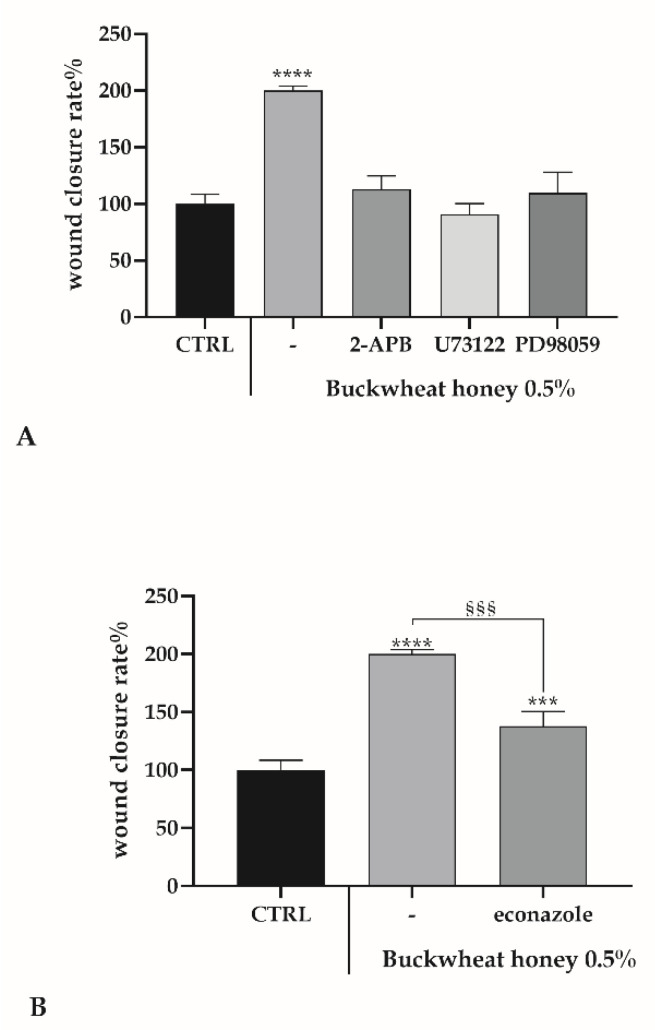
The scratch wound test in bEND5 cells exposed to BH + inhibitors. A wound closure rate after 24 h exposure at 0.5% concentration of BH, in the presence or not of inhibitors. (**A**) Asterisks on bars indicate statistical differences with respect to CTRL determined by a One-Way ANOVA followed by Bonferroni correction (**** *p* < 0.0001). (**B**) Statistics on bars indicate differences with respect to CTRL determined by a One-Way ANOVA followed by Bonferroni correction (*** *p* < 0.001, **** *p* < 0.0001), or differences between BH treatment in the presence or not of econazole determined by a Two-Way ANOVA followed by Tukey’s test (§§§ *p* < 0.001).

**Figure 8 cells-10-01071-f008:**
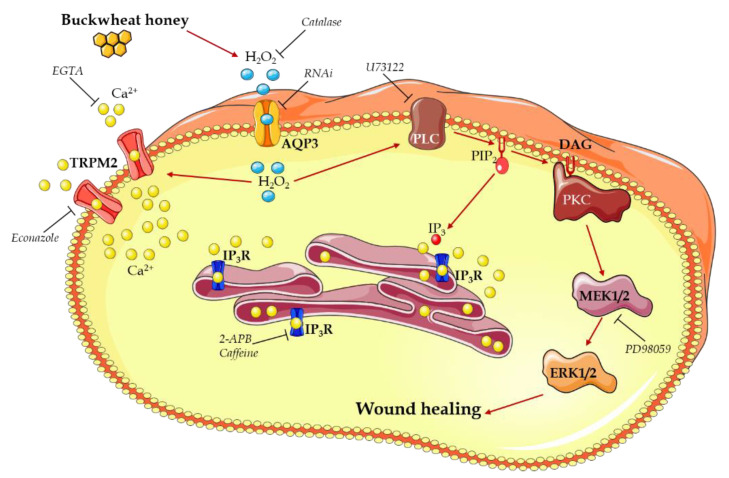
Diagram representing the mechanism of action for buckwheat honey in inducing wound healing in bEND5 endothelial cells. The schematic art pieces used in this figure were provided by Servier Medical art (http://smart.servier.com/, accessed on 19 February 2021). Servier Medical Art by Servier is licensed under a Creative Commons Attribution 3.0 Unported License.

**Table 1 cells-10-01071-t001:** Sequences of siRNA oligonucleotides.

Target Protein	Forward Sequence	Reverse Sequence
*AQP3*	5′-GAGCAGAUCUGAGUGGGCA-3′	5′-UGCCCACUCAGAUCUGCUC-3′

**Table 2 cells-10-01071-t002:** BH effect on cell viability calculated in terms of EC_50_ and EC_05_. Honey concentrations are indicated as % *w*/*v*. The 95% confidence interval is indicated in brackets.

	EC_05_	EC_50_
Buckwheat honey	0.73 (0.35–1.53)	4.72 (3.86–5.76)

## Data Availability

Data are contained within the article or Appendix A.

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
