# Peer review of "Endothelial and Vascular Health: A Tale of Honey, H2O2 and Calcium"

_cells, 2021, doi:10.3390/cells10051071_

Round 1

Reviewer 1 Report

I consider the authors have replied adequately to my questions. 

Reviewer 2 Report

Dear Dr. Simona Martinotti, I have to thank you for your clear and proper answers for reviewer's

opinions. So, I will kindly suggest to Editor to accept your corrected article for the publishing in the journal Cells.

-----------------------------

Reviewer 2  

  Suggestions to Author/s   

1.Dear dr. Simona Martinotti, as a selected reviewer I made a prompt check of your article: »Endothelial and vascular health: a tale of Honey, H2O2 and Calcium«, and found it Excellent. So, I will suggest to the editor, that he will include it in the first acceptable issue of the Journal.   

We thank reviewer for his/her positive comments.  

 2.You are kindly asked, to cite the article: Filipič,B., Gradišnik,L., Ružić-Sabljić, E., Trtnik,B.,Pereyra,A.,Jaklič,D., Kopinč,R.,  Potokar,J., Puzić,A. and Mazija,H. 2016. Water Soluble Propolis and Royal Jelly Enhance the Antimicrobial Activity of Honeysand Promote the Growth of Human Macrophage Cell Line.« Journal of  Agricultural Science and Technology B  6: 35-47. Doi:  

10.17265/2161-264/2016.01.005    

We have inserted this citation in the text.  

   3.Checking your text, the smaller mistakes were found. You are kindly asked to    correct them like this:   

   Line no:     Corrections add or delete  

   ---------------------------------------------------    23       Dell.: an   Add: the   

  1. Add: by   
  1. Add: can   

   63       Add: an   

   90       Add: a   

   96       Add: the   

   Figure 1: Add: the, Add: the, Add: of the   

   135 Add: the   

    158 Add: the   

 Figure 5: Add: the   

 Figure 6: Add: the   

 Figure 7: Add: the; Dell.: Wound; Add: A wound   

297 Add: so   

300 Dell.: the; Add: a; Add: the new   

306 Add: a   

314 Dell.: We; Add: So, we   

317 Add: it was  318 Add: inside   

320 Add: through   

363   Add: intracellular      

We made these corrections.  

Reviewer 3 Report

I thank the authors for addressing the concerns I raised. While a more appropriate approach would have been to include all of the relevant diluent controls and positive controls for the reagents used, I concede that the revised manuscript is significantly improved and suitable for publication.

This manuscript is a resubmission of an earlier submission. The following is a list of the peer review reports and author responses from that submission.

Round 1

Reviewer 1 Report

The paper by Ranzato et al. shows that buckwheat honey (BH) accelerates wound closure in endothelial monolayers by a mechanism involving the extracellular generation of H2O2, H2O2 entry into endothelial cells via aquaporin 3 channels and intracellular triggering by H2O2 of a signaling cascade that induces Ca2+ entry and Ca2+ release. The subsequent increase in intracellular Ca2+ would activate PLC and the MAPK pathway, enhancing wound closure.

Major point:

The paper is well made and adequate evidence is provided for the mechanism described above. However, the authors have already published recently very similar results, with the same type of experiments and the same model but using a culture of keratinocytes. It is true that endothelial repair and new blood vessel formation are very important in wound closing, but the point is that all the mechanisms and model are already published and the main novelty of the article is the use of a different cell preparation.

Other points:

1. There is not a clear correlation between acceleration of wound closure and increase in [Ca2+]. The maximum effect on wound closure is obtained with the minimum concentration of BH used, 0.1%. However, the minimum concentration used to measure the [Ca2+] peak is 0.5%, and there is still a considerable increase in the [Ca2+] peak when increasing BH concentration from 1% to 2%. 0.1% BH should also be used in [Ca2+] experiments, and the differences in the dose-response curves should be explained.

2. If the effect of BH is only mediated by extracellular H2O2 generation, similar effects should be obtained by direct addition or perfusion of a diluted H2O2 solution to the extracellular space. In that way, a direct dose-response curve with H2O2 could be provided. 

Reviewer 2 Report

  • Suggestions to Author/s 

1.Dear dr. Simona Martinotti, as a selected reviewer I made a prompt check of your article: »Endothelial and vascular health: a  tale of Honey,  H2O2 and Calcium«, and  found it Excellent. So, I will suggest to the editor, that he  will include it in the first acceptable issue of the Journal. 

 2.You are kindly asked, to cite the  article: Filipič,B., Gradišnik,L., Ružić-Sabljić, E., Trtnik,B.,Pereyra,A.,Jaklič,D., Kopinč,R.,  Potokar,J., Puzić,A. and Mazija,H. 2016. »Water Soluble Propolis and Royal Jelly Enhance the Antimicrobial Activity of Honeysand Promote the Growth of Human Macrophage Cell Line.« Journal of  Agricultural Science and Technology B  6: 35-47. Doi: 10.17265/2161-264/2016.01.005  

   3.Checking your text, the smaller mistakes were found. You are kindly asked to    correct them like this: 

   Line no:     Corrections add or dellete 

   ---------------------------------------------------

   23       Dell.: an   Add: the 

   24       Add: by 

   25       Add: can 

   63       Add: an 

   90       Add: a 

   96       Add: the 

   Figure 1: Add: the, Add: the, Add: of the 

   135 Add: the 

    158 Add: the 

 Figure 5: Add: the 

 Figure 6: Add: the 

 Figure 7: Add: the ; Dell.: Wound; Add: A wound 

297 Add: so 

300 Dell.: the ; Add: a; Add: the new 

306 Add: a 

314 Dell.: We ; Add: So, we 

317 Add: it was 

318 Add: inside 

320 Add: through 

363   Add: intracellular    

Reviewer 3 Report

In this study, Ranzato et al aim to determine the capacity of buckwheat honey (genus Fagopyrum), a polyphenol-rich dark honey known for its anti-inflammatory properties to stimulate wound closure in endothelial cells, investigating more in depth the signal cascade activation.

In my opinion, the data presented by Ranzato et al has been inadequately controlled and thereby over interpreted by the authors. While this study addresses some interesting points, I have concerns related to the data presented and have listed them below for consideration.

Major comments:

  1. English grammar needs extensive checking and correcting.
  2. What was the non-targeting control sequence for the RNAi studies? Why are these control RNAi experiments not included in the studies?
  3. Where did the blossom honey come from? Why is it not used more extensively in the study? Are blossom honey and buckwheat honey the same thing? Or is this a typo?
  4. Figure 1 needs to include images from time 0 for all experimental conditions. The scratches do not look even.
  5. What is the control diluent for the econazole? Why is this not included?
  6. Because many endothelial cell lines differ in their protein expression, the authors should show that TRPM2 is indeed expressed by the bEND5 cells.
  7. The authors should show evidence that the inhibitors work as expected, eg a documented readout for exonazole on the TRPM2, 2-APB for the IP3-induced Ca2+ release and U73122 for the PLC. None of these are included in the submitted manuscript.
  8. Data should be included to support their claim of PD98059 inhibition of the MEK1/2 in their cells.